# Some mechanistic underpinnings of molecular adaptations of SARS-COV-2 spike protein by integrating candidate adaptive polymorphisms with protein dynamics

**Nicholas James Ose[1], Paul Campitelli[1], Tushar Modi[1], I Can Kazan[1], Sudhir Kumar[2,3,4]\*, Sefika Banu Ozkan[1]\***

[1]Department of Physics and Center for Biological Physics, Arizona State University, Tempe, United States; [2]Institute for Genomics and Evolutionary Medicine, Temple University, Philadelphia, United States; [3]Department of Biology, Temple University, Philadelphia, United States; [4]Center for Genomic Medicine Research, King Abdulaziz University, Jeddah, Saudi Arabia

**\*For correspondence:**
s.kumar@temple.edu (SK);
banu.ozkan@asu.edu (SBO)

**Competing interest:** The authors declare that no competing interests exist.

**Abstract** We integrate evolutionary predictions based on the neutral theory of molecular evolution with protein dynamics to generate mechanistic insight into the molecular adaptations of the SARS-COV-2 spike (S) protein. With this approach, we first identified candidate adaptive polymorphisms (CAPs) of the SARS-CoV-2 S protein and assessed the impact of these CAPs through dynamics analysis. Not only have we found that CAPs frequently overlap with well-known functional sites, but also, using several different dynamics-based metrics, we reveal the critical allosteric interplay between SARS-CoV-2 CAPs and the S protein binding sites with the human ACE2 (hACE2) protein. CAPs interact far differently with the hACE2 binding site residues in the open conformation of the S protein compared to the closed form. In particular, the CAP sites control the dynamics of binding residues in the open state, suggesting an allosteric control of hACE2 binding. We also explored the characteristic mutations of different SARS-CoV-2 strains to find dynamic hallmarks and potential effects of future mutations. Our analyses reveal that Delta strain-specific variants have non-additive (i.e., epistatic) interactions with CAP sites, whereas the less pathogenic Omicron strains have mostly additive mutations. Finally, our dynamics-based analysis suggests that the novel mutations observed in the Omicron strain epistatically interact with the CAP sites to help escape antibody binding.

## eLife assessment

This **important** study investigates various variants of the SARS-COV-2 spike protein using established computational methods, complemented by experimental validation efforts. The evidence, bolstered by an evolutionary approach and protein dynamics, is **solid**. Placing this research in the broader context of the field could further enrich the article. It will interest biophysicists focused on allostery and protein evolution.

## Introduction

Since 2019, the evolution of SARS-CoV-2 in humans has been characterized by the spread of mutations, many notably found within the spike (S) glycoprotein. The S protein is directly related to the human immune response to COVID-19 and, as such, has been one of the most studied and targeted

proteins in SARS-CoV-2 research (*Shang et al., 2020*; *Harvey et al., 2021*; *Jackson et al., 2022*; *Carabelli et al., 2023*; *Markov et al., 2023*). Subsequently, research into the biophysical properties and mutational patterns associated with S protein evolution not only remains critical to understanding the pandemic but also emerges as a useful system to understand the mechanics of molecular adaptation within viruses.

For successful infection of a human host, the S glycoprotein of SARS-CoV-2 binds to the human ACE2 (hACE2) receptor through its receptor-binding domain (RBD). Evidence indicates that fine-tuning S protein interactions with hACE2 significantly affects viral reproduction (*Rehman et al., 2020*; *Saputri et al., 2020*; *Rochman et al., 2021*). Previous evolutionary studies show a complex network of interactions among mutated residues (*Changeux and Edelstein, 2005*; *Doshi et al., 2016*; *O'Rourke et al., 2016*; *Mishra and Jernigan, 2018*). Therefore, there has been a vast effort to uncover which mutations are important steps of adaptation for the S protein (*Cagliani et al., 2020*; *Damas et al., 2020*; *Singh and Yi, 2021*; *Kistler et al., 2022*; *Maher et al., 2022*; *Neher, 2022*). In particular, a significant aspect of many such studies was a focus on understanding adaptive mutations of SARS-CoV-2 that contributed to the leap to human hosts (*Cagliani et al., 2020*; *Damas et al., 2020*; *Singh and Yi, 2021*; *Starr et al., 2022c*). This is because SARS-CoV-2 has continuously mutated since its early detection (*Kistler et al., 2022*), causing the emergence of CDC-designated 'variants of concern' (VOCs) that may be driven by an accelerated substitution rate (*Tay et al., 2022*).

Predicting how new mutations impact the biophysical properties of the S protein remains a challenge, let alone explaining their complex interactions with one another and how they might affect hACE2 binding; many factors affect hACE2 interactions. Binding affinity with hACE2 can be enhanced directly through stronger receptor interactions or mediated through changes in RBD opening (*Teruel et al., 2021b*; *Zhang et al., 2021*; *Díaz-Salinas et al., 2022*). The RBD exhibits both 'closed' and 'open' conformational states. In the closed state, the RBD is shielded from receptor binding. In the open state, the RBD is accessible for hACE2 binding (*Kirchdoerfer et al., 2016*; *Gur et al., 2020*; *Henderson et al., 2020*; *Hoffmann et al., 2020*). While some mutations may affect the transition between these states (*Henderson et al., 2020*; *Yurkovetskiy et al., 2020*; *Gobeil et al., 2021a*; *Sztain et al., 2021*; *Zhang et al., 2021*; *Shoemark et al., 2022*), other mutations may allosterically regulate RBD openings through Furin cleavage site (residue ID range: 681–695) interactions to regulate hACE2 binding (*Deng et al., 2021*; *Gobeil et al., 2021b*; *Khan et al., 2021*; *Laiton-Donato et al., 2021*).

Moreover, as new mutations accumulate, culminating in the emergence of a new VOC, these mutations must occur on varied sequence backgrounds containing neutral, nearly-neutral, and adaptive mutations. While many studies have explored the impacts of individual mutations, VOCs result in a substantial difference in protein function compared to their individual effects (*Moulana et al., 2022*; *Starr et al., 2022c*; *Moulana et al., 2023*; *Witte et al., 2023*). Here we integrate an evolutionary approach with protein dynamics analysis to address the fundamental mechanisms of mutations dictating VOCs and the impact of their epistatic interaction on the function of the S protein. Many earlier studies have combined phylogeny and evolutionary theory to identify adaptive mutations and analyze how the viral sequence has changed over time (*Frost et al., 2018*; *Boni et al., 2020*; *Cagliani et al., 2020*; *Damas et al., 2020*; *Tang et al., 2020*). Similarly, we first use a well-established evolutionary probability (EP) approach (*Liu et al., 2016*) that utilizes phylogenetic trees in combination with the neutral theory of molecular evolution to determine candidate adaptive polymorphisms (CAPs) using the early Wuhan sequence as a variant. CAPs are substitutions in SARS-CoV-2 that are rarely observed in other closely related sequences (*Figure 1A*), which implies a degree of functional importance and makes them candidates for adaptation (*Liu et al., 2016*). Adaptation in this case means a virus which can successfully infect human hosts. As CAPs are unexpected polymorphisms under neutral theory, their existence implies a non-neutral effect. This can come in the form of functional changes (*Liu et al., 2016*) or compensation for functional changes (*Ose et al., 2022b*). Therefore, we suspect that these CAPs may be partially responsible for the functional change allowing the infection of human hosts. In support of this method, we find an overlap between our list of sites containing CAPs and putative adaptive sites identified by others (*Cagliani et al., 2020*; *Singh and Yi, 2021*; *Kistler et al., 2022*; *Starr et al., 2022b*). Second, we use a suite of computational tools to analyze how CAPs that arose in the early and late phases of the COVID-19 pandemic modulate the dynamics of the S protein. We also explore the complex interactions between these sets of CAPs to gain mechanistic

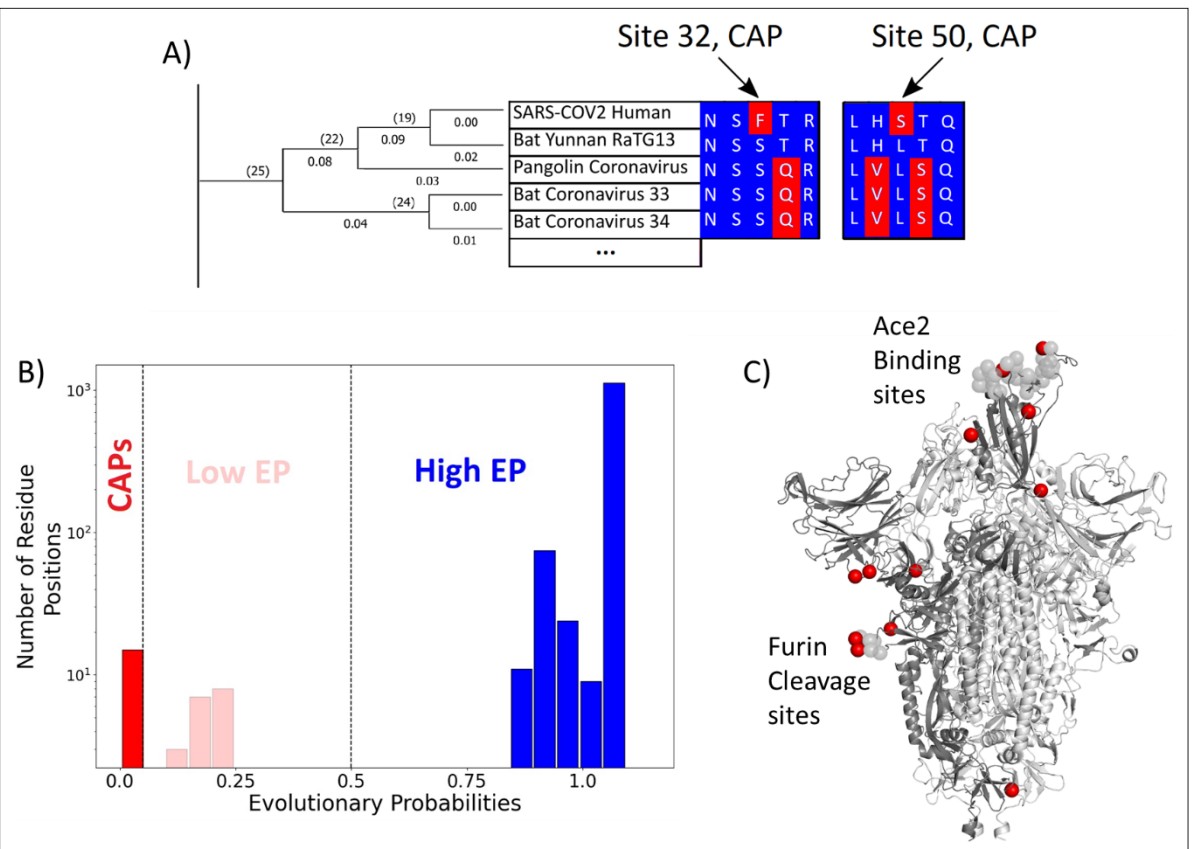

**Figure 1.** Evolutionary probability (EP) in SARS-CoV-2. (**A**) The EPs of each amino acid in the S protein sequence are calculated by taking the multiple sequence alignment of the S proteins through their evolutionary tree and using Bayesian inferences to determine the likelihood of finding a particular residue at a particular location within a given sequence. Simply, if the residue is found at a location 'x' in closely related sequences, it will have a higher EP at location 'x' in the target sequence. Residues with an EP < 0.05 in the target sequence are candidate adaptive polymorphisms (CAPs) (red). CAPs are found at sites 32, 50, 218, 346, 372, 478, 486, 498, 519, 604, 681, 682, 683, 684, and 1125. (**B**) The distribution of EP scores of the wild-type residues in the S protein. Here, lower EP scores are shown in red, and higher EP scores in blue. While the vast majority of the wild-type (reference) protein consists of high EP residues, a few residues have low EP. (**C**) The CAPs are also highlighted as red spheres in the open configuration of the S protein, with the open chain in a darker shade. We observe that a majority of the CAP positions reside at the receptor-binding domain (RBD) and the Furin cleavage site (676-689; *Wrobel et al., 2020*) shown as transparent light gray spheres.

insight into the behavior of molecular adaptation involving the S protein. In particular, we focused on how mutations modulate protein dynamics as we and others have previously found that rather than changing a protein's structure solely, mutations modulate conformational dynamics, leading to changes in biophysical properties such as stability, flexibility, and allosteric dynamic coupling, any of which may affect protein binding (*Swint-Kruse et al., 1998*; *Keskin et al., 2000*; *Bhabha et al., 2013*; *Nussinov and Tsai, 2013*; *Campbell et al., 2016*; *Ma and Nussinov, 2016*; *Saavedra et al., 2018*; *Kuzmanic et al., 2020*).

With this evolutionary-dynamics unified approach, we aim to answer the following questions about VOCs: Are all the mutations in VOCs adaptive in nature? Are they coupled to one another or provide some measure of biophysical, dynamical, or mechanical epistasis? While many of these mutations are found within the RBD, numerous others are located distal to this region; hence, we aim to investigate the functional roles of distal mutations, particularly from a protein dynamics perspective. Our integrated analysis revealed that protein dynamics play a significant role in the evolution of the S protein. The flexibility of sites within the S protein shows a strong, direct correlation with substitution rate, and newly evolving CAPS are mostly additive mutations that modulate the dynamics of the hACE2 binding site. Yet other CAPs, 346R, 486F, and 498Q, show highly epistatic (i.e., non-additive) modulation of the hACE2 binding site and provide immune escape benefits.

## Results and discussion

### Candidate adaptive mutations in the S protein are recognized via EPs

SARS-CoV-2 is part of a family of coronaviruses, many of which infect mainly animals and are less capable of infecting humans (*Dicken et al., 2021*). Therefore, to identify the most likely mutations responsible for the infection of human hosts (i.e., putative adaptive mutations for humans), we estimated the (neutral) EP scores of mutations found within the S protein (*Liu et al., 2016*). EP scores of the amino acid variants of the S protein were obtained using a maximum likelihood phylogeny (*Kumar et al., 2018*) built from 19 orthologous coronavirus sequences. Sequences were selected by examining available non-human sequences with a sequence identity of 70% or above to the human SARS CoV-2's S protein sequence. This cutoff allows for divergence over evolutionary history such that each amino acid position had ample time to experience purifying selection, whilst limiting ourselves to closely related coronaviruses. (*Figure 1A*). The likelihood of finding a particular amino acid in the sequence is then determined using a Bayesian framework, with calculations carried out by MEGA X software (*Kumar et al., 2018*). As apparent in the name, EP scores obtained for the amino acids in the sequence provide information regarding the likelihood of finding them at their position, given the history of the sequence. Amino acid residues receiving low EP scores (<0.05) at a position are less likely to be found in a given position within the sequence because they are non-neutral. Generally, positions with low EP amino acids are far less common than those containing mutations with high EP, a trend also realized in the CoV-2 S protein (*Figure 1B*).

Of particular interest is an observed evolutionary change where an amino acid with high EP is replaced by an amino acid residue with low EP. While amino acids with low EP should be harmful or deleterious to viral fitness due to functional disruption or change, fixation of a low EP amino acid at a position suggests an underlying mechanism for natural selection to operate. These fixed, low EP mutations are called CAPs as they are predicted to alter protein function, and adaptive pressures may drive their prevalence (*Kumar and Patel, 2018*). Indeed, there is an overlap between these CAPs and the mutations suggested by other methods to be adaptive for the S protein (*Cagliani et al., 2020*; *Singh and Yi, 2021*; *Kistler et al., 2022*; *Starr et al., 2022b*). Within these studies, sites 478, 486, 498, and 681 have been implicated in SARS-CoV-2 evolution, leaving the remaining 11 CAPs as undiscovered candidate sites for adaptation.

Interestingly, most of the CAP residues are at functionally critical sites, including the RBD and the Furin cleavage site (*Figure 1C*). As mentioned earlier, the RBD plays a key role in initiating the infection of a healthy cell by binding with the host organism's ACE2 protein. Before ACE2 binding, one chain of the homotrimer comprising the S protein must open to expose the RBD (*Kirchdoerfer et al., 2016*; *Henderson et al., 2020*; *Hoffmann et al., 2020*; *Sztain et al., 2021*). The Furin cleavage site plays a key role in the opening process as the binding of host cell protease Furin aids in the cleavage of the S protein into two domains: S1 and S2 (*Wrobel et al., 2020*: 13). Another host cell protease, TMPRSS2, facilitates viral attachment to the surface of target cells upon binding either to sites Arg815/Ser816, or Arg685/Ser686 which overlap with the Furin cleavage site 676–689, further emphasizing the importance of this area (; *Fraser et al., 2022*). Similar cleavage sites have been found in related coronaviruses, including HKU1 and Middle East respiratory syndrome coronavirus (MERS-CoV), which infects humans (*Chan et al., 2008*: 1; *Millet and Whittaker, 2014*; *Millet and Whittaker, 2015*), and the acquisition of similar cleavage sites is associated with increased pathogenicity in other viruses such as the influenza virus (*Steinhauer, 1999*). Interestingly, however, CAPs do not display such an overwhelming tendency to occur at well-known critical sites within human proteins studied with similar methods (*Ose et al., 2022b*), yet mutations at those sites are associated with disease, indicating their critical role in inducing functional change. Therefore, the identified CAPs in the S protein, which are signs of recent evolution, can provide mechanistic insights regarding the molecular adaptation of the virus. In particular, we aimed to analyze how these CAP positions in the S protein modulate the interaction with hACE2 using our protein dynamics-based analysis (*Gerek and Ozkan, 2011*; *Nevin Gerek et al., 2013*; *Larrimore et al., 2017*; *Kumar et al., 2015*).

### Asymmetry in communications among the network of interactions in spike describes how CAPs regulate the dynamics of the S protein

A mutation at a given amino acid position inevitably not only alters local interactions, but this change cascades through the residue–residue interaction network, which gives rise to a variation in native

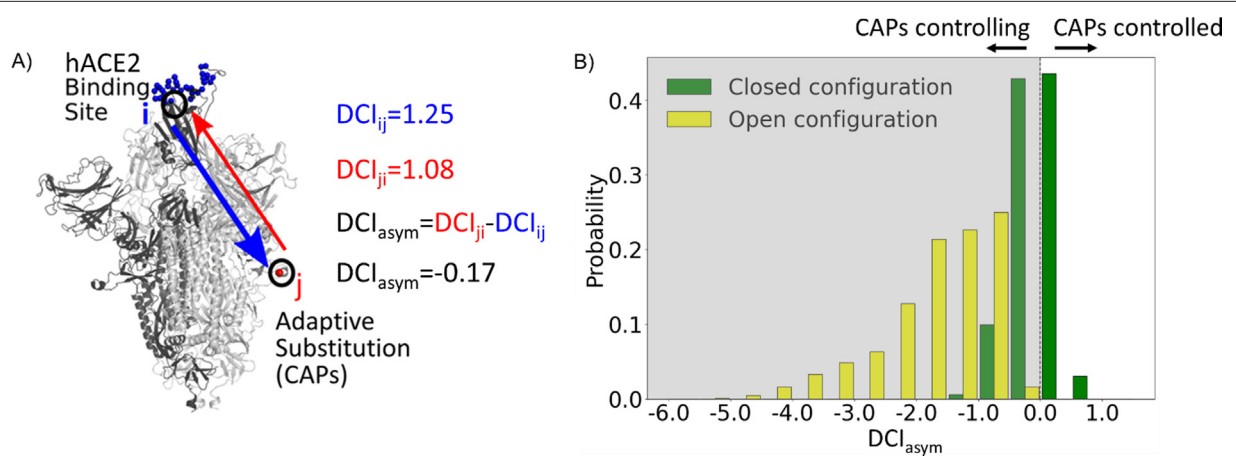

**Figure 2.** Candidate adaptive polymorphisms (CAPs) control binding sites across chains. (**A**) Schematic representation of DCI$_{asym}$. (**B**) DCI$_{asym}$ of CAP residue positions with the binding interface of receptor-binding domain (RBD) in the open chain. Residues in the closed chains with a low evolutionary probability (EP) amino acid in the reference sequence dominate the binding site interface of RBD in the open chain. There is a significant difference between the asymmetry profiles of the closed (M = –0.06, SD = 0.33) and open (M = –1.68, SD = 0.89) conformations (p<0.001).

ensemble dynamics to modulate function (*Dror et al., 2012*; *Labbadia and Morimoto, 2015*; *Sekhar and Kay, 2019*; *Campitelli et al., 2020*.). Many groups have already examined the conformational dynamics of the S protein using normal mode analysis to explore mutation sites and interactions with different receptors (*Zhou et al., 2020*; *Majumder et al., 2021*; *Teruel et al., 2021a*; *Verkhivker, 2022*). However, our study will focus mainly on the role of CAPs, of which F486 and Q498 have already been identified through perturbation response scanning (PRS) as potential allosteric sites (*Verkhivker, 2022*). Thus, we analyze the internal dynamics of the system to understand the functional role of CAPs in S proteins. This analysis allows us to gain a mechanistic understanding of the relationship between CAP mutations and biophysical outcomes (*Teruel et al., 2021b*). First, we implement the dynamic coupling index (DCI) approach to study long-distance coupling between the CAPs and the hACE2 binding sites emerging from the 3D network of interactions across the S protein system. DCI calculation combines PRS and linear response theory (LRT) to capture the strength of a displacement response for position *i* upon perturbation of position *j*, relative to the average fluctuation response of position *i* to all other positions in the protein. It represents the strength of dynamic coupling between positions *i* and *j* upon perturbation to *j* (*Larrimore et al., 2017*; *Kumar et al., 2015*).

Further, asymmetry can be captured in the DCI values as dynamic coupling is not necessarily symmetric due to an anisotropic interaction network. That is, each amino acid has a set of positions to which it is highly coupled, and this anisotropy in connections gives rise to unique differences in coupling between a given *i*, *j* pair of amino acids which do not have direct interactions. By calculating the coupling of the hACE2 binding interface in the RBD with respect to the CAP residue positions and vice versa, we can generate DCI$_{asym}$ (*Figure 2A*) as the difference between the normalized displacement response of position *j* upon a perturbation to position *i* (DCI$_{ij}$) and the normalized displacement response of position *i* upon a perturbation to position *j* (DCI$_{ji}$) (see 'Methods'). If the DCI$_{asym}$ values significantly differ from zero, it shows asymmetry in coupling and presents a cause–effect relationship between the *i*, *j* pair in terms of force/signal propagation. This metric has been used previously in a variety of systems to analyze the unique behavior of positions within a protein and a given position's propensity to effect biophysical changes upon mutation, particularly at long distances (*Modi and Ozkan, 2018*; *Campitelli and Ozkan, 2020*; *Kolbaba-Kartchner et al., 2021*; *Ose et al., 2022a*; *Kazan et al., 2023*; *Campitelli et al., 2021*).

Recent work from our group has shown an enhancement in cross-chain communication within the main protease of SARS COV-2 compared to SARS COV-1 (*Campitelli et al., 2022*). Furthermore, previous studies have shown that allosteric inter-chain communication is important to S protein function (*Zhou et al., 2020*; *Spinello et al., 2021*; *Tan et al., 2022*; *Xue et al., 2022*). In support of these findings, we observe through DCI$_{asym}$ that when the S protein is in its pre-fusion conformation with one chain open, the CAPs in the closed chains have negative coupling asymmetry with respect to the

hACE2 binding site interface in the RBD-open chain. This indicates an allosteric control where the hACE2 binding site is dominated by the dynamics of the CAPs in closed chains (*Figure 2B*, yellow bars). As this open-state RBD is critical for the viral infection of host cells (*Kirchdoerfer et al., 2016*), our results suggest that this type of closed-to-open cross-chain interaction is important for viral proliferation. Our prior studies on $DCI_{asym}$ show a similar trend in lactose inhibitor (LacI), a protein with a functional role in gene expression through binding DNA. The allosteric mutations (i.e., mutations on the sites that are far from the DNA binding sites) that alter DNA binding affinity not only exhibited unique asymmetry profiles with the DNA binding sites of LacI, but also regulated the dynamics of these binding sites (*Campitelli et al., 2021*).

Similarly, it is possible that mutations to such residue positions within the S protein can be used to regulate the dynamics of the hACE2 bindings sites of the open RBD state. We, therefore, propose that the residue positions with CAP substitutions hold the potential for mutations in the spike sequence which can alter the opening and closing dynamics of the RBD. This hypothesis is further supported by many mutations already observed at these residue positions which alter the infection rate (*Brister et al., 2015*). Interestingly, residues responsible for extremely low asymmetry values (<–4) lie overwhelmingly in the region 476–486. These same residues were suggested to stabilize S protein dynamics and prime it for host Furin proteolysis (*Raghuvamsi et al., 2021*).

Moreover, as a control, we performed the same analysis on the S protein with the RBDs of all chains in the closed configuration. In this case, we observed that the $DCI_{asym}$ of the CAPs residue positions with respect to the hACE2 interface in the other chains yields a largely symmetric distribution about 0 (*Figure 2B*, green bars). This verifies that the asymmetry in the coupling of CAPs with the exposed binding site interface in pre-fusion configuration results from one of the RBDs opening up and further suggests the allosteric role played by CAPs in locking the S protein in the RBD open state.

## Dynamics analysis shows that rigid sites tend to be more highly conserved than flexible sites

CAPs represent important S protein amino acid changes between related coronaviruses across multiple species and the Wuhan-Hu-1 reference sequence (MN908947). Since SARS-CoV-2 first spread to

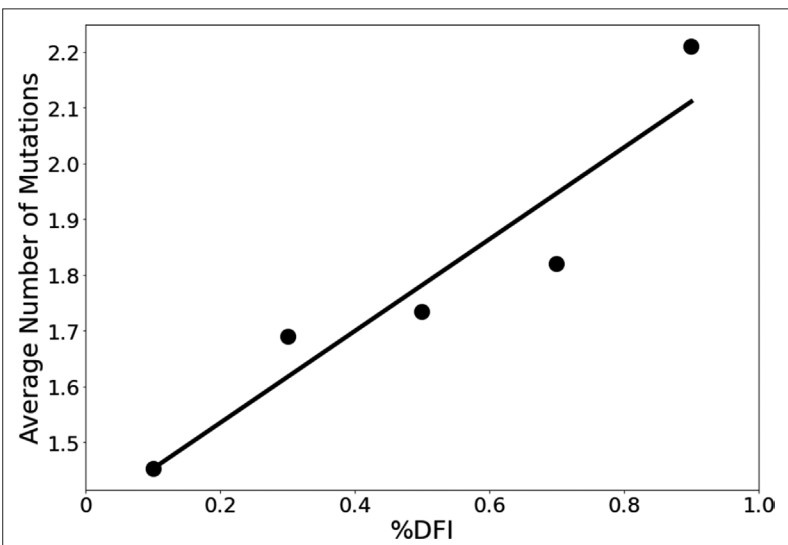

**Figure 3.** The average number of variants observed among residues of different flexibility. Residues were sorted into one of five bins based on flexibility. Then, the average number of variants for residues within that bin was calculated. Here, the number of variants is defined as the number of different amino acid varieties found at that site. Mutational data was calculated across approximately 24,000 SARS-CoV-2 S protein sequences from the NCBI Datasets Project (*Brister et al., 2015*). Residue flexibility, as reported here via %DFI, was computed using PDB id 6vsb from the Protein DataBank (*Berman et al., 2000*). More rigid residues tend to have fewer variants (*r* = 0.94).

The online version of this article includes the following source data for figure 3:

**Source data 1.** A comparison of the amino acid sequences for different variant spike proteins used in this article.

humans, it has continued to mutate and evolve rapidly, particularly regarding the S protein (*Amicone et al., 2022*; *Liu et al., 2022*; *Tay et al., 2022*). Just as mutations leading to the Wuhan strain caused an increase in binding affinity to hACE2, continued evolution in human hosts has resulted in further altered binding affinities as well as different phenotypic outcomes for those infected (*Ali et al., 2021*; *Barton et al., 2021*; *Ozono et al., 2021*).

We explore whether protein dynamics has played a role in the selection of mutational sites during the evolution of the S protein since 2019. Our previous work has indicated that the rate of evolution per positional site exhibits a positive correlation with positional flexibility; generally, positions that exhibit higher flexibility are also sites that experience a higher number of amino acid substitutions (*Liu and Bahar, 2012*; *Maguid et al., 2008*; *Maguid et al., 2006*; *Mikulska-Ruminska et al., 2019*; *Nevin Gerek et al., 2013*). To confirm these findings for the evolution of the S protein using the sequenced variants of infected humans, we examine the flexibility of the S protein, an analysis conducted in other studies which resulted in several important findings (*Nguyen et al., 2020*; *Socher et al., 2021*; *Teruel et al., 2021b*; *Pipitò et al., 2022*; *Verkhivker, 2022*; *Abduljalil et al., 2023*). For example, *Teruel et al., 2021a* used their Elastic Network Contact Model to find how certain highly observed mutations make the open state more rigid and the closed state more flexible. For our own flexibility analysis, we measure the site-specific amino acid flexibility using the dynamic flexibility index (DFI). Using the same mathematical foundation as DCI, DFI evaluates each position's displacement response to random force perturbations at other locations in the protein (*Gerek and Ozkan, 2011*; *Nevin Gerek et al., 2013*), and it can be considered a measure of a given position's ability to explore its local conformational space. We found that the COVID-19 S protein shows the expected high correlation between the occurrence of mutations and site flexibility (*Figure 3*) when we compare %DFI (DFI ranked by percentile) to the average number of variants per position found within a given %DFI bin. Previous studies have indicated that rigid residues are critical for functional dynamics, thus more likely to impact function if mutated and, generally, can lead to a loss of function and thus more conserved (*Kim et al., 2015*; *Butler et al., 2018*; *Modi et al., 2021b*; *Modi et al., 2021a*; *Kazan et al., 2022*; *Ose et al., 2022a*; *Stevens et al., 2022*; *Kumar et al., 2015*). This analysis also agrees with these previous studies and highlights the power of negative selection, in line with the neutral theory of molecular evolution, stating that deleterious mutations (i.e., those on the rigid positions) should be eliminated and therefore not observed (*Kimura, 1983*).

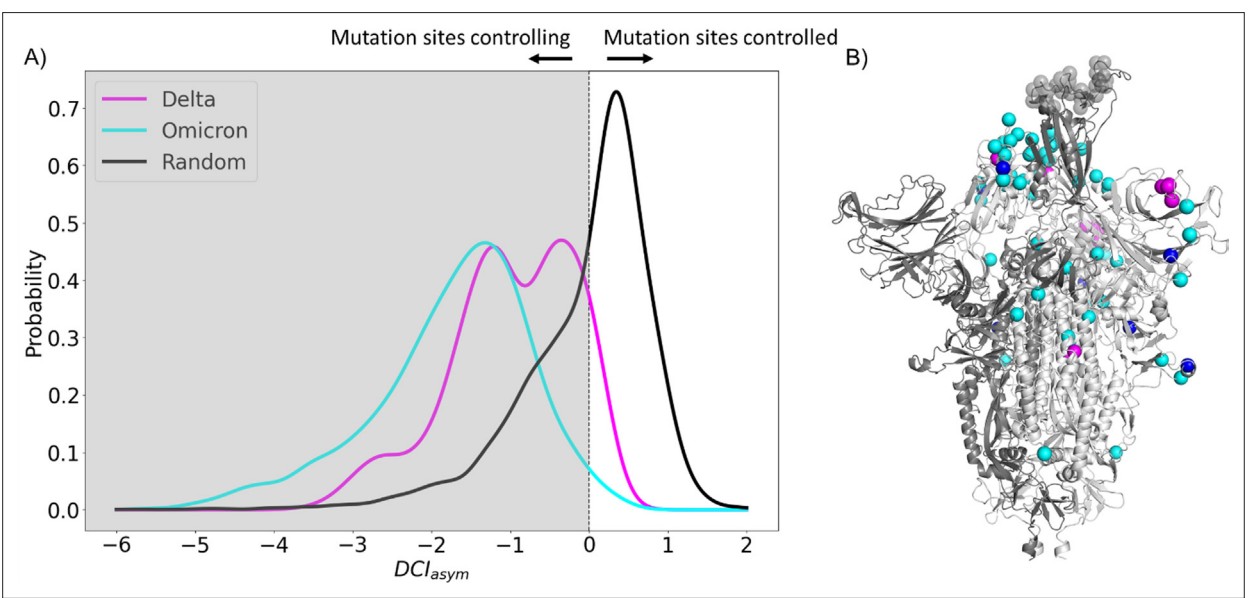

**Figure 4.** Mutations in variants of concern (VOCs) present different asymmetry profiles. (**A**) DCI$_{asym}$ with low evolutionary probability (EP) characteristic mutation sites of Delta or Omicron strains in the closed chains and the binding interface of receptor-binding domain (RBD) in the open chain. Delta displays a second peak closer to zero, suggesting that Delta mutation sites (M = –0.98, SD = 0.80) have less allosteric control over the hACE2 binding sites than Omicron mutation sites (M = –1.74, SD = 1.00) (p<0.001). However, both sets of sites have far more control over hACE2 binding sites than expected, based on a random control group (M = 0.03, SD = 0.85) (p<0.001). (**B**) S protein structure showing binding interface sites (transparent gray), Delta mutation sites (magenta), Omicron mutation sites (cyan), and sites mutated in both Omicron and Delta (blue).

Continued mutations within human hosts have resulted in a multitude of variants. Indeed, by fitting various molecular clock models to genome sequence data, VOC emergence is punctuated by an episodic period of rapid evolution, with a substitution rate of up to fourfold greater than the background substitution rate (*Kumar et al., 2021*; *Tay et al., 2022*). With such an aggressive evolutionary rate, we are finding VOCs to consist of a number of different characteristic mutations, almost all of which are CAPs. Sequence differences between the various VOCs used in this article can be found in *Figure 3—source data 1*.

To explore the dynamic effects of the evolution of the spike in humans, we examine asymmetry with these new potentially adaptive sites, namely the low EP (CAP) characteristic mutation sites observed in the Delta variant, the widely dominant variant from December 2021 to January 2022 (*Thye et al., 2021*), and the Omicron variant, a highly transmissible variant whose lineages have remained dominant since January 2022 (*Kim et al., 2021*; *Figure 4*). This analysis revealed a mechanism similar to that for the CAPs in the reference protein (*Figure 2*) as the open-chain binding interface is also allosterically controlled by these potentially new adaptive sites. Regarding this, we see that the asymmetry is much more pronounced in observed mutations of Omicron variants, suggesting that these new mutations have a stronger power in controlling the dynamics of open-chain hACE2 binding interface compared to those observed in Delta variants. We surmise that the difference in virulence and infection rates between Omicron and Delta (*Earnest et al., 2022*; *Bager et al., 2021*; *Sheikh et al., 2021*; *Twohig et al., 2022*; *Houhamdi et al., 2022*; *Menni et al., 2022*) might be due to these specific CAPs within each variant and their differences in allosterically controlling the dynamics of open RBD binding sites as also observed in the DCI$_{asym}$ analysis of the Wuhan variant in *Figure 2*.

## Experimental results motivate the use of EpiScore within the SARS-CoV-2 S protein

The fact that the identified CAPs in the reference protein and the more recently evolved CAPs of Delta and Omicron variants both show a high degree of control over the functional sites begs the question: what is the complex interaction between these previous and new CAP sites? Motivated by this concept, we explore the interplay of mutational pairs to understand the effects of the specific amino acid backgrounds associated with these two predominant variants. Some CAP sites in Delta and Omicron have already been considered adaptive (*Kemp et al., 2021*; *Kistler et al., 2022*; *Maher et al., 2022*; *Neher, 2022*).

It is well understood that the impact of even a single mutation to a protein sequence can sometimes dramatically alter the biophysical behavior of the system. However, the mechanistic impact of point mutations can only be fully understood when the sequence background upon which it is made is accounted for. This means that, in the case of strains with multiple mutations, the interplay between mutated positions will ultimately impact a protein as an aggregate behavior, where the presence of previous mutations may strongly (or weakly) influence some mutations. This concept of non-additivity is known as epistasis. In fact, studies of evolutionary pathways of mutations have suggested that a majority of the mutations have a second or a higher order epistasis among them (*Bershtein et al., 2006*). Nature exploits this higher order complex relationship between the mutations to evolve their function.

To computationally capture and interpret the pairwise effects of mutations, we have developed an in-house computational tool called EpiScore (*Figure 5A*). Here, we evaluate how a given position pair $i$ $j$ may affect other critical positions $k$ of the protein. EpiScore is the relative coupling strength to a position $k$ when positions $i$ and $j$ are perturbed *simultaneously* compared to the average dynamic coupling strength of $i$ to $k$ and $j$ to $k$. EpiScore has previously been used successfully to capture overarching trends in GB1 deep mutational scan data as well as specific instances of the development of antibiotic resistance in various enzymatic systems (*Campitelli and Ozkan, 2020*). An EpiScore of 1 indicates perfect coupling additivity, and deviations from this value represent non-additive behavior between position pairs and functionally important sites. Prior EpiScore work has shown a difference in EpiScore between the sites of compensatory and non-compensatory mutations, where both yield distributions with peaks around 1, but non-compensatory mutations show higher deviation in their EpiScore distribution (*Ose et al., 2022b*).

Many studies have confirmed epistasis between residues within the S protein (*Moulana et al., 2022*; *Starr et al., 2022a*; *Moulana et al., 2023*; *Witte et al., 2023*). These epistatic residues can have various

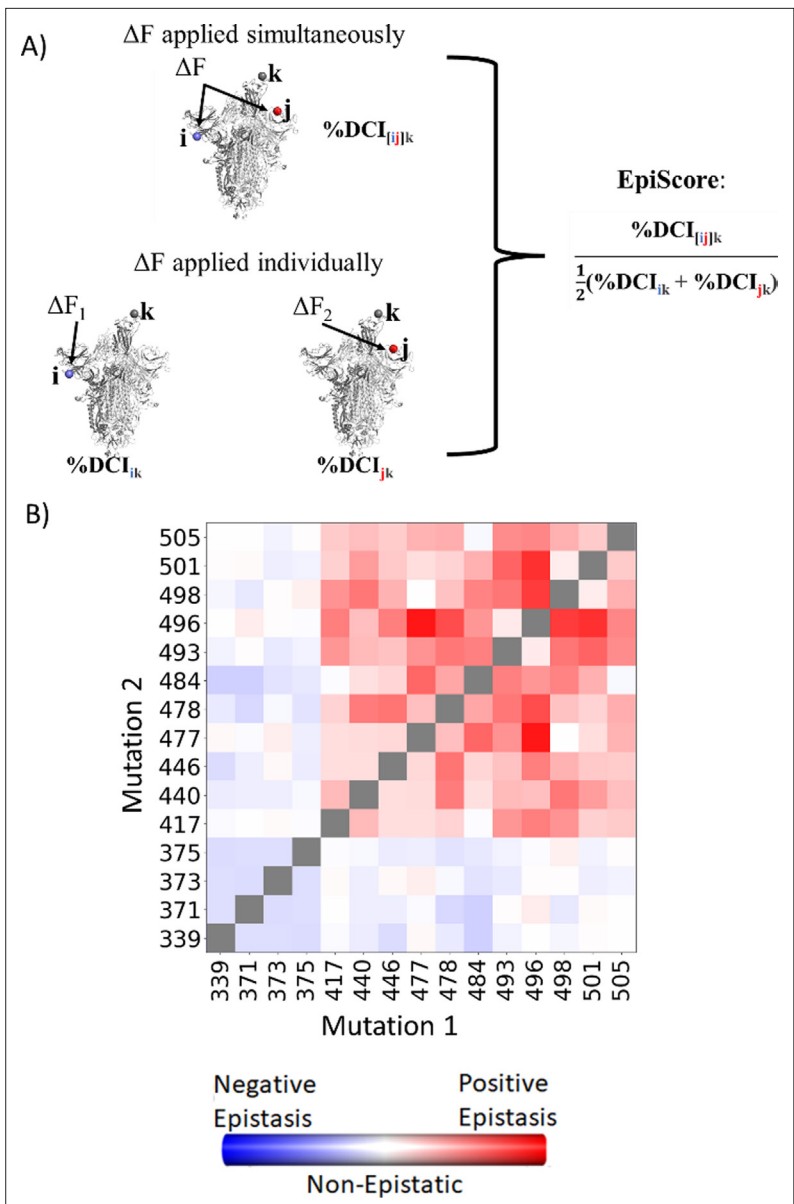

**Figure 5.** EpiScore provides a measurement of epistasis. (**A**) Schematic representation of cross-chain EpiScore, describing *i, j*, in chains B and C, respectively, and its impact in receptor-binding domain (RBD) binding position k in the open RBD conformer chain A. (**B**) Colors indicate EpiScore values for given mutation pairs, averaged over hACE2 binding sites. Cross-chain residue pairs in the upper right tend to be highly epistatic, similar to pairwise second-order interaction coefficients from *Moulana et al., 2022*.

effects on hACE2 or antibody binding. Within SARS-CoV-2, here we calculate the EpiScore (*Figure 5B*) of a set of mutation pairs used by *Moulana et al., 2022* and compare our results to quantified epistatic effects determined by the experimental hACE2 binding affinity of 'first-order' single mutation variants compared to 'second-order' mutation pairs. Our EpiScore results and the experimentally determined epistasis both captured highly epistatic behavior among residues 493, 496, 498, 501, and 505, as well as a lack of epistatic behavior for residues 339, 371, 373, and 375; however, EpiScore generally showed higher epistasis values than experiment for residues 417, 440, 446, 477, 478, and 484.

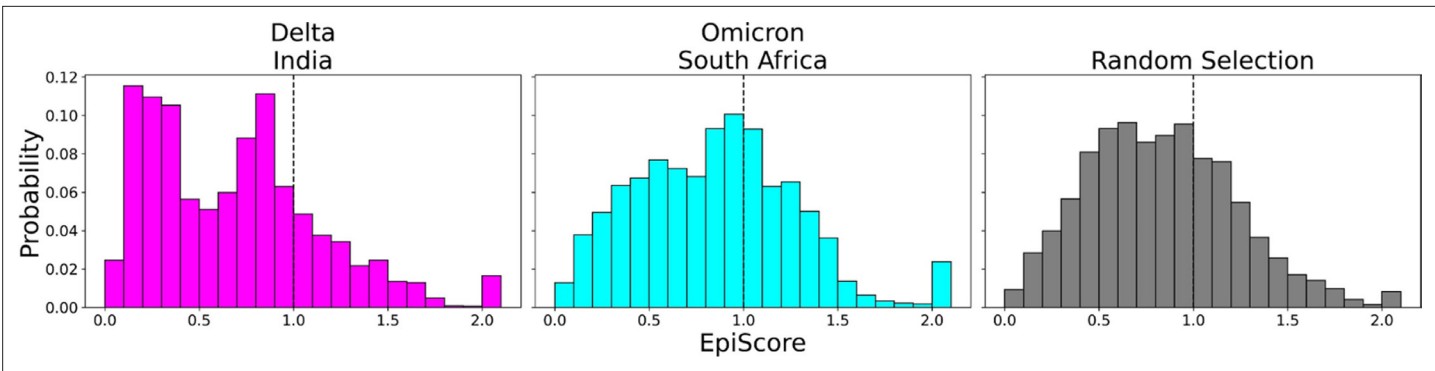

**Figure 6.** EpiScores of variants of concern (VOC) mutations. Here, *i* = low evolutionary probability (EP) Delta mutation sites (magenta), low EP Omicron mutation sites (cyan), and a random selection of sites (gray), *j* = low EP sites in the Wuhan variant, and *k* = the binding interface of the open chain. EpiScores using sites of either variant (Delta: M = 0.70, SD = 0.50, Omicron M = 0.86, SD = 0.46) are significantly different (p<0.001) from a set of EpiScores using random sites (M = 0.84, SD = 0.41), but the distribution for Delta variants differs much more from the other two. EpiScores for other variants can be found in *Figure 6—figure supplement 1*.

The online version of this article includes the following figure supplement(s) for figure 6:

**Figure supplement 1.** EpiScores for more variants of concern (VOCs).

**Figure supplement 2.** EpiScores in the N-terminal domain (NTD).

## EpiScore highlights the epistatic relationship between the recent adaptive mutations in VOCs and the CAPs of the Wuhan reference

Seeking further to understand the role of epistasis within S protein variants, we explored the possibility of epistatic relationships between the CAPs of the Wuhan variant and the new CAPs in VOCs. Thus, we computed the EpiScore of these CAP positions in the closed RBDs (i.e., chains B and C) with respect to functional hACE2 binding interface sites of the open RBD chain (chain A) (*Figure 5B*) and obtained EpiScore distributions.

To contrast these variants, Omicron (*Figure 6*, cyan) shows a high proportion of additive mutations compared to the Delta variant (*Figure 6*, magenta), with a peak centered on 1. The comparatively more pathogenic Delta variant exhibited many non-additive mutations with EpiScores below one. This again suggests that the cross-communication between the open and closed chain of the S protein is important for regulating the function. Four out of seven low EP Delta mutation sites used in this analysis often resulted in EpiScores below 1. Each of those is found in the N-terminal domain (NTD) on or near the N3 loop and is implicated in antibody escape in recent studies (*Chi et al., 2020*; *Weisblum et al., 2020*; *Harvey et al., 2021*; *Klinakis et al., 2021*; *Cantoni et al., 2022*). The low EpiScores of NTD mutations suggest that they dampen the control of Wuhan variant CAPs over the hACE2 binding sites in addition to their effects on antibody binding. It is possible that what the Delta variant gained in transmission rate also came with being more harmfully pathogenic due in part to negatively epistatic interactions. In contrast, the mutations leading to the development of the Omicron strain were additive with respect to Wuhan variant CAPs, possibly leading to a lower pathogenicity and higher effective immune escape, resulting in an overall higher transmission rate. Another possible explanation for the higher transmission rate of Omicron comes from a different normal mode analysis study, which found that despite reduced binding affinity with hACE2, Omicron showed increased occupancy of the open state compared to the closed state, which would increase chances of interaction with hACE2 (*Teruel et al., 2021b*). It is worth noting that other variants contain NTD mutations which result in low EpiScores; however, the proportion of these mutations within the set is considerably less than in Delta (*Figure 6—figure supplement 2*).

One of the more notable features of generated EpiScore distributions is the presence of a tail of values upward of 2.0, indicating highly epistatic behavior. Interestingly, these tails are largely due to three different CAPs: 346R, 486F, and 498Q. Those residues are nearby one another within the protein structure and have been reported to play a role in antibody binding, either being known antibody binding sites (346R and 486F) or having received very high antibody accessibility scores (498Q) (*Harvey et al., 2021*; *Raghuvamsi et al., 2021*). These observed high EpiScore values also support

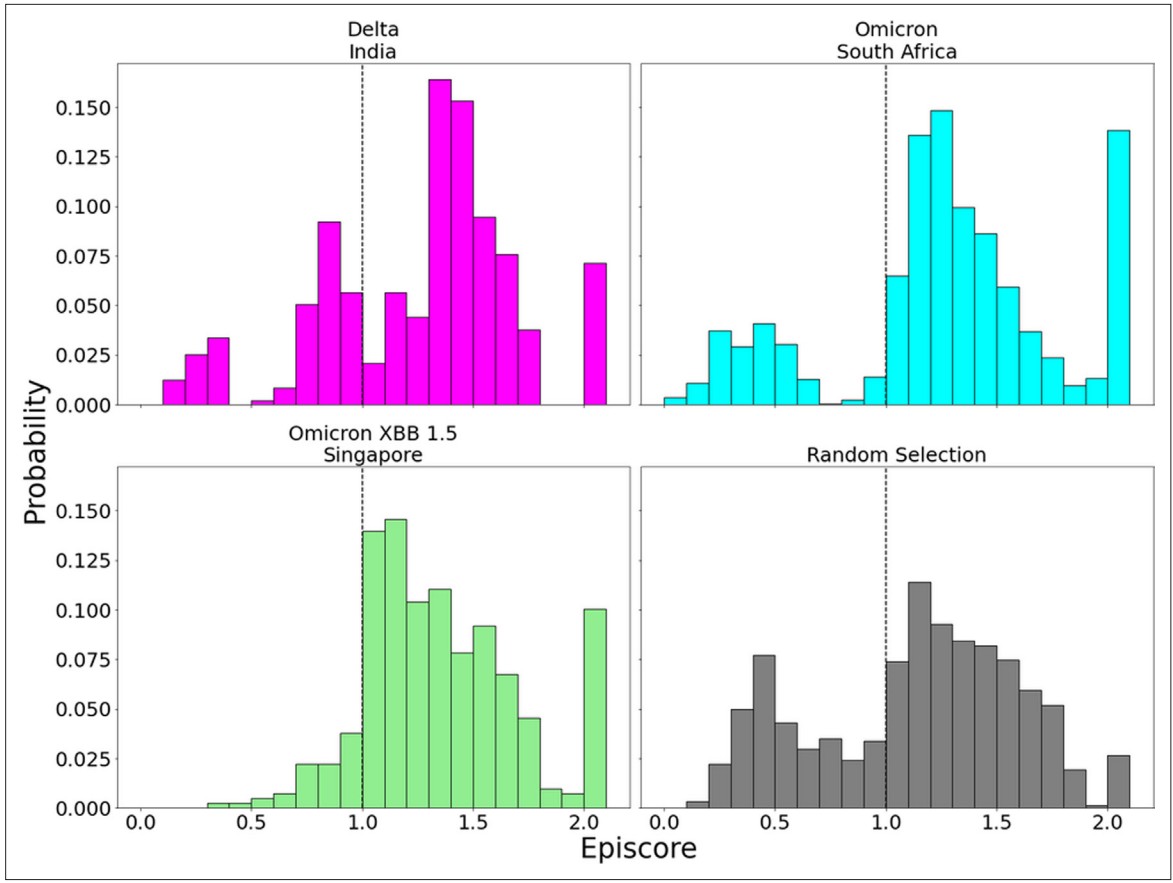

**Figure 7.** EpiScores with candidate adaptive polymorphism (CAP) site 486. Here, *i* = low evolutionary probability (EP) Delta mutation sites (magenta), low EP Omicron mutation sites (cyan), and a random selection of sites (gray), *j* = site 486, and *k* = the binding interface of the open chain. CAP and hACE2 and antibody binding site 486 displays epistasis with almost all XBB 1.5 variant sites at almost every hACE2 binding site (M = 1.40, SD = 0.46) and presents a significantly different profile from other variant sites (p<0.01). EpiScores involving 486 for other variants can be found in *Figure 7—figure supplement 1*.

The online version of this article includes the following figure supplement(s) for figure 7:

**Figure supplement 1.** EpiScores for more variants of concern (VOCs) with candidate adaptive polymorphism (CAP) site 486.

other studies indicating the epistatic interactions between these CAPs and the mutations of the VOCs within the S protein are crucial for maintaining binding affinity of hACE2 whilst evading immunity (*Hong et al., 2022*; *Starr et al., 2022b*).

Inspection of EpiScores of Delta and Omicron potentially adaptive mutation sites with only CAP site 486F (a binding site for both hACE2 and antibodies) (*Huang et al., 2020*; *Ali et al., 2021*; *Harvey et al., 2021*; *Raghuvamsi et al., 2021*) shows highly epistatic interactions at other hACE2 binding sites (*Figure 6*). However, within a recent and rapidly spreading subvariant of Omicron, XBB 1.5, we see a mutation of S to P, a rare double nucleotide mutation, at site 486 (preceding Omicron variants included mutation F486S from the original CAP). This new variant has unprecedented immune escape capabilities, resisting neutralizing antibodies almost entirely (*Qu et al., 2023*). EpiScores of other XBB 1.5-specific mutation sites with 486P are almost entirely greater than 1, showing an even higher degree of epistasis with the binding sites of RBD (*Figure 7*). These results present a threefold importance for the S486P mutation: not only does this residue alter antibody (i.e., immune escape) and hACE2 binding by directly modifying a binding site, but it may also be responsible for modifying hACE2 binding via epistatic cooperation with other co-occurring mutations.

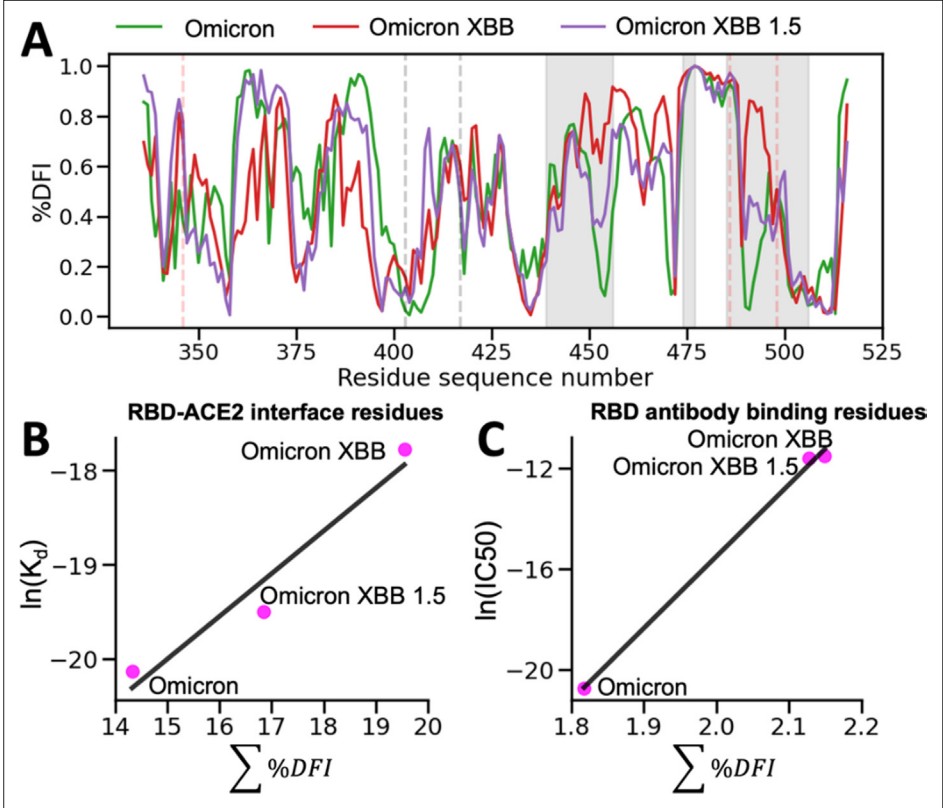

**Figure 8.** The %DFI calculations for variants Omicron, XBB, and XBB 1.5. (**A**) %DFI profile of the variants are plotted in the same panel. The gray shaded areas and dashed lines indicate the ACE2 binding regions, whereas the red dashed lines show the antibody binding residues. (**B**) The sum of %DFI values of RBD-ACE2 interface residues. The trend of total %DFI with the log of $K_d$ values overlaps with the one seen with the experiments ($r$ = 0.97). (**C**) The receptor-binding domain (RBD) antibody binding residues are used to calculate the sum of %DFI. The ranking captured with the total %DFI agrees with the log of IC50 values from the experiments.

The online version of this article includes the following figure supplement(s) for figure 8:

**Figure supplement 1.** Non-additive interactions influence variant behavior.

## Change in flexibility of RBD binding site correlates with experimental binding affinities for Omicron and Omicron XBB variants

Experimental studies have tracked hACE2 binding for different variants since the virus first spread (*Ali et al., 2021*; *Barton et al., 2021*; *Ozono et al., 2021*; *Wu et al., 2022*). Within the Omicron variant, for example, characteristic mutations on the RBD are shown to increase the overall binding affinity of the virus to the ACE2 receptor, which is suspected to allow it to spread more easily (*Kim et al., 2021*). Furthermore, the new Omicron XBB and Omicron XBB 1.5 variants contain additional mutations in the RBD and antibody binding residues, which may further impact their dynamics and interactions with the host.

To gain deeper insights into the impact of dynamics on the binding affinity of hACE2 and antibodies with the recent Omicron XBB variants, we conducted molecular dynamics (MD) simulations. By analyzing the resulting trajectories, we investigated how these mutations influence the flexibility and rigidity of the RBD and antibody binding residues, consequently affecting their binding affinity and potential for immune evasion (*Figure 8*). To understand the overall flexibility changes, we measured the sum of DFI of the ACE2 binding residues, as well as the sum of DFI of the antibody binding residues, calculated from the MD trajectories and compared them with experimental viral binding (disassociation constants) and immunity evasion antibody IC50 values (*Yue et al., 2023*).

This investigation elucidated the impact of mutations in the RBD and antibody binding residues on the binding affinity of the S protein and immune evasion by modulating their flexibility and rigidity (*Figure 8A*). The Omicron XBB variant exhibits heightened flexibility in hACE2 and antibody

binding residues, reducing infectivity and enhancing immune evasion. Conversely, the Omicron XBB 1.5 variant induces distinct dynamics in these regions, rendering the RBD-ACE2 interface more rigid while increasing flexibility in antibody binding residues. These effects indicate that Omicron XBB 1.5 retains its antibody escape capabilities while regaining ACE2 binding affinity comparable to previous Omicron variants, in accordance with experimental findings (*Yue et al., 2023*). These findings suggest that mutations in the RBD and antibody binding residues can have complex effects on the dynamics of the protein and, ultimately, on the virus's ability to infect and evade the host immune system through an alteration of binding site dynamics. However, we do note that the effect of mutations will heavily depend on the genetic background in which they are mutated. The effects of a mutation on a Delta variant protein may differ entirely from the effects of a mutation on an Omicron variant protein (*Figure 8—figure supplement 1*).

## Conclusion

We analyzed the evolutionary trajectory of the CoV-2 S protein in humans to understand the dynamic and epistatic interactions of the mutations defining specific VOCs. We first obtain the phylogenetic tree of the COV-2 S protein and identify the sites of certain recent mutations known as CAPs. CAPs are considered adaptive because mutations rarely tolerated in closely related sequences have suddenly become fixed, implying a degree of functional importance or evolution (*Liu et al., 2016*). In addition, our earlier work has shown that CAPS can also be compensatory; multiple CAPs may dynamically compensate for one another, changing the dynamic landscape and allowing for different mutations (*Ose et al., 2022b*). We then explored the mechanistic insights and epistatic relationship between the observed mutations in different VOCs and CAP sites, and, particularly, the relationship between CAP sites and the functionally critical RBD using our dynamic coupling analysis (*Kumar et al., 2015*).

We find a mechanistic pattern in the S protein evolution that is common amongst previously studied systems, where allosteric sites exert control over the dynamics of the binding sites, and mutations of these allosteric sites modulate function. Coupling our analysis with evolutionary theory showed that many of these allosteric sites regulating function of the S protein may have been subjected to adaptive evolution as observed in mutations in VOCs. Our dynamics analysis also provides a mechanistic insight where the Omicron-defining sites have greater control over the binding sites than the Delta variant and are dynamically additive with the functional advantage of CAPs, thus, the greatest infectivity may not be a coincidence.

Specifically, we find that the interactions between CAP sites and VOC-defining mutations show fingerprints of non-additive dynamics within the Delta variant. In contrast, mutations leading to the Omicron variant are largely additive, driving critical dynamical behavior closer to the patterns observed within the wild-type. These interactions may also drive observed behavior similar between the reference and Omicron strains yet differ in the delta strain, such as the severity of infection as evidenced by hospitalization rates (*Houhamdi et al., 2022*; *Menni et al., 2022*). It has also been shown that the Omicron variant has a lower binding affinity with hACE2 than previous variants (*Wu et al., 2022*), which may contribute to its low pathogenicity.

Long-ranged interactions between different sites within a given protein are critically important for protein function (*Peters and Lively, 1999*; *Bershtein et al., 2006*; *Collins et al., 2006*; *Ekeberg et al., 2013*; *Levy et al., 2017*; *Harrigan et al., 2018*; *Otten et al., 2018*; *Rojas Echenique et al., 2019*; *Shimagaki and Weigt, 2019*; *de la Paz et al., 2020*; *Rizzato et al., 2020*; *Yang et al., 2020*; *Bisardi et al., 2022*) and for the CoV-2 S protein in particular (*Zeng et al., 2020*; *Castiglione et al., 2021*; *Dong et al., 2021*; *Garvin et al., 2021*; *Nielsen et al., 2022*; *Ramarao-Milne et al., 2022*; *Rochman et al., 2022*; *Rodriguez-Rivas et al., 2022*). By showing dynamic differences between the interactions of CAPs, which have likely played a major role in allowing the virus to infect human hosts, the binding site, and the characteristic mutations of dominant Delta and Omicron strains, we see a 'fine-tuning' of protein behavior. As variants continue to evolve, Omicron subvariants are of growing concern due in large part to further increased immune evasion (*Callaway, 2022*; *Wang et al., 2022a*; *Wang et al., 2022b*), and we observe that the new mutations observed in antibody binding sites yield more epistatic interaction with the CAPs. In addition to supporting previous dynamic research on the S protein, this analysis provides the insight that CAP sites are of continued importance to protein function and should be given special attention when considering the impact of future mutations.

## Methods

### Dynamic flexibility and dynamic coupling

The DFI utilizes a PRS technique that combines the elastic network model (ENM) and LRT (*Gerek and Ozkan, 2011*; *Nevin Gerek et al., 2013*). In ENM, the protein is considered as a network of beads at Cα positions interacting with each other via a harmonic spring potential. Using LRT, Δ**R** is calculated as the fluctuation response vector of residue $j$ due to unit force's **F** perturbation on residue $i$, averaged over multiple unit force directions to simulate an isotropic perturbation.

$$[\Delta R]_{3N \times 1} = [H]^{-1}_{3N \times 3N} [F]_{3N \times 1} \tag{1}$$

where **H** is the Hessian, a 3N × 3N matrix that can be constructed from 3-D atomic coordinate information and is composed of the second derivatives of the harmonic potential with respect to the components of the position's vectors of length 3N. The Hessian inverse in this equation may be replaced with the covariance matrix **G** obtained from MD simulations as follows:

$$[\Delta R]_{3N \times 1} = [G]_{3N \times 3N} [F]_{3N \times 1} \tag{2}$$

MD simulations were used to obtain the DFI profiles of Omicron, Omicron XBB, and Omicron XBB 1.5. In order to obtain DFI, each position in the structure was perturbed sequentially to generate a Perturbation Response Matrix **A**

$$A_{N \times N} = \begin{bmatrix} \left|\Delta R^1\right|_1 & \cdots & \left|\Delta R^N\right|_1 \\ \vdots & \ddots & \vdots \\ \left|\Delta R^1\right|_N & \cdots & \left|\Delta R^N\right|_N \end{bmatrix} \tag{3}$$

where $\left|\Delta R^j\right|_i = \sqrt{\left\langle (\Delta R)^2 \right\rangle}$ is the magnitude of fluctuation response at position $i$ due to perturbations at position $j$. The DFI value of position $i$ is then treated as the displacement response of position $i$ relative to the net displacement response of the entire protein, which is calculated by sequentially perturbing each position in the structure.

$$DFI_i = \frac{\sum_{j=1}^{N} \left|\Delta R^j\right|_i}{\sum_{i=1}^{N} \sum_{j=1}^{N} \left|\Delta R^j\right|_i} \tag{4}$$

It is also often useful to quantify position flexibility relative to the flexibility ranges unique to individual structures. To that end, DFI can be presented as a percentile rank, %DFI. All %DFI calculations present in this work used the DFI value of every residue of the full spike structure for ranking. The DFI parameter can be considered a measure of a given amino acid position's ability to explore its local conformational space.

### Dynamic coupling index

Similar to DFI, the DCI (*Larrimore et al., 2017*; *Kumar et al., 2015*) also utilizes PRS with the ENM and LRT. DCI captures the strength of displacement response of a given position $i$ upon perturbation to a single functionally important position (or subset of positions) $j$, relative to the average fluctuation response of position $i$ when all of the positions within a structure are perturbed.

$$DCI_{ji} = \frac{\sum_{j}^{N_{functional}} \left|\Delta R^j\right|_i / N_{functional}}{\sum_{j=1}^{N} \left|\Delta R^j\right|_i / N} \tag{5}$$

When only positional pairs are concerned, this expression reduces to

$$DCI_{ji} = \frac{\left|\Delta R^j\right|_i}{\sum_{j=1}^{N} \left|\Delta R^j\right|_i / N} \tag{6}$$

As such, this parameter represents a measure of the dynamic coupling between $i$ and $j$ upon a perturbation to $j$. As with DFI, $DCI_{ji}$ can also be presented as a percentile-ranked $\%DCI_{ji}$.

One of the most important aspects of DCI is that the entire network of interactions is explicitly included in subsequent calculations without the need for dimensionality reduction techniques. If one considers interactions such as communication directionality or dynamic coupling regulation between position pairs as inherent properties of an anisotropic interaction network, it is critical to include the interactions of the entire network to accurately model the effect one residue can have on another.

Here, we present two further extensions of DCI, which allow us to uniquely model coupling directionality and epistatic effects: $DCI_{asym}$ and EpiScore, respectively. Interestingly, we can capture asymmetry between different residues within a protein through DCI, as a coupling in and of itself is asymmetric within an anisotropic network. That is, each amino acid has a set of positions to which it is highly coupled, and this anisotropy in connections gives rise to unique differences in coupling between a given $i\,j$ pair of amino acids which do not have direct interactions (*Figure 2A*). $DCI_{asym}$, then, is simply $DCI_{ij}$ (the normalized displacement response of position $j$ upon a perturbation to position $i$) − $DCI_{ji}$ (*Equation (7)*). Using $DCI_{asym}$, we can determine a cause–effect relationship between the $i\,j$ pair in terms of force/signal propagation between these two positions.

$$DCI_{asym} = DCI_{ij} - DCI_{ji} \tag{7}$$

$$\%DCI_{asym} = \%DCI_{ij} - \%DCI_{ji} \tag{8}$$

where a positive $DCI_{asym}$ value indicates communication from position $i$ to position $j$.

EpiScore can identify or describe potential non-additivity in substitution behavior between residue pairs. This metric can capture the differences in a normalized perturbation response to a position $k$ when a force is applied at two residues $i$ and $j$ simultaneously versus the average additive perturbation response when each residue $i$, $j$, is perturbed individually (*Figure 5A*, *Equation 9*).

$$EpiScore = \frac{\%DCI_{[ij]k}}{\frac{1}{2}\left(\%DCI_{ik} + \%DCI_{jk}\right)} \tag{9}$$

EpiScore values <1 (>1) indicate that the additive perturbations of positions $i$ and $j$ generate a greater (lesser) response at position $k$ than the effect of a simultaneous perturbation. This means that, when treated with a simultaneous perturbation at both sites $i$ and $j$, the displacement response of $k$ is lower (higher) than the average effect of individual perturbations to $i$ and $j$, one at a time. As EpiScore is a linear scale, the further the value from 1, the greater the effect described above.

## Molecular dynamics (MD)

The production simulations for the Omicron variants, including Omicron, Omicron XBB, and Omicron XBB 1.5, were performed with the AMBER software package. These variants, each characterized by specific mutations, were modeled based on the template PDB structure 6M0J. The initial protein configurations in the simulations were parameterized using the ff14SB force field (*Maier et al., 2015*). In order to create an appropriate solvation environment for the proteins, a solvation box was defined around them, maintaining a minimum separation distance of 16 Å from the protein to the box boundaries. This was accomplished by employing the explicit TIP3P water model (*Sun and Kollman, 1995*), with the addition of sodium and chloride ions to maintain overall charge neutrality.

The simulation procedure involves an initial energy minimization step, aimed at mitigating steric clashes and optimizing the system's energy. The steepest descent algorithm was applied, encompassing 11,000 steps. Subsequently, the system underwent a gradual temperature increase (heat up), up to 300 K, and was subjected to production simulations under a constant number of particles, pressure, and temperature ensemble (NPT).

During these production simulations, temperature was maintained at 300 K, with pressure regulation set at 1 bar. Temperature regulation was achieved through the utilization of the Langevin thermostat (*Hünenberger, 2005*) and Berendsen barostat (*Berendsen et al., 1984*), featuring a collision frequency of 1.0 picoseconds⁻¹. Hydrogen atom bond lengths were constrained using the SHAKE algorithm (*Pearlman et al., 1995*). The production trajectories were simulated for 1 μs each.

To ensure the reliability of the simulations and assess their convergence, a convergence criterion was employed. The achieved convergence was determined by monitoring the root mean square

deviation (RMSD) between the highest sampled conformation in consecutive time windows (*Sawle and Ghosh, 2016*). Specifically, convergence was defined as the point at which the RMSD between the highest sampled conformation in the last 300 ns window and the 300 ns window immediately preceding it dropped below 1 Å. Window sizes varying from 100 ns to 500 ns were employed to evaluate convergence, ensuring the robustness and stability of the obtained results.

To calculate DFI, covariance matrix data were computed over different time windows as discussed above. By default, utilizing the Hessian implies a restriction to a harmonic potential, assuming that the data are sampled from a Gaussian distribution. Ergodicity in both simulation time and initial structures sampled in each time interval ensures two key conditions: (i) consistency of potential energy across conformations sampled from the same distribution. (ii) The sampling of different initial conformations while computing covariance matrices at various time windows eliminates global motions and accurately captures equilibrium coordinates. Consequently, the final average DFI profiles are independent of time window size, resulting in consistent results across different time window sizes (e.g., 50 ns vs. 75 ns) and enabling the acquisition of statistically significant DFI values.

### Statistical tests

Pearson correlation coefficients ($r$) were used to demonstrate linear relationships between continuous variables in *Figures 3 and 8*. Student's independent *t*-tests were performed to demonstrate a significant difference between distributions in *Figures 2, 4 and 6* (and *Figure 3—source data 1*, *Figure 6—figure supplements 1 and 2*), and *Figure 7* (and *Figure 7—figure supplement 1*), as demonstrated by the p-value.

### Acknowledgements

Funding was provided to NJO, PC, TM, and ICK by the Gordon and Betty Moore Foundation (award number AWD00034439) and to SBO by the National Science Foundation (award numbers: 1715591 and 1901709) and the National Institutes of Health R01GM147635-01. SK acknowledges the National Science Foundation (GCR 1934848) and the National Institutes of Health (GM139540) grants.

## Additional information

### Funding

| Funder | Grant reference number | Author |
| --- | --- | --- |
| Gordon and Betty Moore Foundation | AWD00034439 | Nicholas James Ose<br>Paul Campitelli<br>Tushar Modi<br>I Can Kazan |
| National Science Foundation | 1715591 | Sefika Banu Ozkan |
| National Science Foundation | 1901709 | Sefika Banu Ozkan |
| National Science Foundation | GCR 1934848 | Sudhir Kumar |
| National Institutes of Health | R01GM147635-01 | Sefika Banu Ozkan |
| National Institutes of Health | GM139540 | Sudhir Kumar |

The funders had no role in study design, data collection and interpretation, or the decision to submit the work for publication.

### Author contributions

Nicholas James Ose, Conceptualization, Data curation, Software, Formal analysis, Investigation, Methodology, Writing – original draft, Writing – review and editing; Paul Campitelli, Conceptualization,

Software, Formal analysis, Methodology, Writing – original draft, Writing – review and editing; Tushar Modi, Conceptualization, Software, Formal analysis, Methodology, Writing – review and editing; I Can Kazan, Conceptualization, Software, Formal analysis, Investigation, Methodology, Writing – review and editing; Sudhir Kumar, Conceptualization, Data curation, Software, Supervision, Funding acquisition, Methodology, Writing – review and editing; Sefika Banu Ozkan, Conceptualization, Supervision, Visualization, Methodology, Writing – original draft, Project administration, Writing – review and editing

### Author ORCIDs
Nicholas James Ose  http://orcid.org/0000-0002-2194-5199
Paul Campitelli  http://orcid.org/0000-0001-5620-609X
Tushar Modi  http://orcid.org/0000-0001-9483-9170
I Can Kazan  https://orcid.org/0000-0003-2593-4179
Sudhir Kumar  http://orcid.org/0000-0002-9918-8212
Sefika Banu Ozkan  https://orcid.org/0000-0002-9351-3758

Reviewer #2 (Public review): https://doi.org/10.7554/eLife.92063.3.sa1
Reviewer #3 (Public review): https://doi.org/10.7554/eLife.92063.3.sa2
Author response https://doi.org/10.7554/eLife.92063.3.sa3

## Additional files

### Supplementary files
- Supplementary file 1. DFI profile of the Omicron variant, as obtained from MD simulations.
- Supplementary file 2. DFI profile of the Omicron XBB variant, as obtained from MD simulations.
- Supplementary file 3. DFI profile of the Omicron XBB 1.5variant, as obtained from MD simulations.
- Supplementary file 4. Mutation sites and EP values.
- Supplementary file 5. The alignment used to generate EP values.
- Supplementary file 6. Alignment of VOC sequences.
- MDAR checklist

### Data availability
The code to perform DFI and DCI analysis is available at https://github.com/SBOZKAN/DFI-DCI (copy archived at *Ozkan, 2024*). Molecular Dynamics data are available at https://github.com/SBOZKAN/COV2SPIKE_MD. The mutation sites and EP values are contained in *Supplementary file 4*. The alignment used to generate EP values is also contained within the supporting information files as "EP_alignment.fas". Protein Databank ID number 6VXX (*Walls et al., 2020*) was used for closed conformation DCI calculations. 6VSB (*Wrapp et al., 2020*) was used for DFI calculations, EpiScore calculations, and open conformation DCI calculations. 6M0J (*Lan et al., 2020*) was used in molecular dynamics simulations of the RBD.

The following dataset was generated:

| Author(s) | Year | Dataset title | Dataset URL | Database and Identifier |
|---|---|---|---|---|
| Ozkan SB | 2024 | COV2SPIKE_MD | https://github.com/SBOZKAN/COV2SPIKE_MD | GitHub, 91b9d69 |

The following previously published datasets were used:

| Author(s) | Year | Dataset title | Dataset URL | Database and Identifier |
|---|---|---|---|---|
| Wrapp D, Wang N, Corbett KS, Goldsmith JA, Hsieh C, Abiona O, Graham BS, McLellan JS | 2020 | Prefusion 2019-nCoV spike glycoprotein with a single receptor-binding domain up | https://www.rcsb.org/structure/6vsb | RCSB Protein Data Bank, 6VSB |
| Wang X, Lan J, Ge J, Yu J, Shan S | 2020 | Crystal structure of SARS-CoV-2 spike receptor-binding domain bound with ACE2 | https://www.rcsb.org/structure/6m0j | RCSB Protein Data Bank, 6M0J |
| Walls AC, Park YJ, Tortorici MA, Wall A | 2020 | Structure of the SARS-CoV-2 spike glycoprotein (closed state) | https://www.rcsb.org/structure/6vxx | RCSB Protein Data Bank, 6VXX |

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
