## [Editor Report · eLife assessment]

This **important** study investigates various variants of the SARS-COV-2 spike protein using established computational methods, complemented by experimental validation efforts. The evidence, bolstered by an evolutionary approach and protein dynamics, is **solid**. Placing this research in the broader context of the field could further enrich the article. It will interest biophysicists focused on allostery and protein evolution.

---

## [Referee Report · Reviewer #2 (Public review)]

The authors set out to identify CAPs (Candidate Adaptive Polymorphyisms), i.e., simply put mutations that carry a potential functional advantage, and utilize computational methods based on the perturbation of C-alpha positions with an Elastic Network Model to determine if dynamics of CAP residues are different in any way.

The authors have addressed the main methodological concerns.

However, one point remains. The specific comparison of which CAPs have been previously identified by other means, particularly with other computational methods that look into dynamics is still lacking. It is unfortunate that the authors do not present such analysis, particularly with respect to single point mutational analysis of Teruel et al. in Plos Comp. Bio. If CAP positions were previously identified by other means it adds strength to the methodology used by the authors. The authors also do not discuss their results in light of the work of Lam et al. (Sci. Comm, 2020) where an evolutionary analysis of Spike/ACE2 binding across homologues is performed. I believe that such deeper discussion of the current results in light of previous work, adds strength to the analysis presented in this manuscript as the methodology is different. Even if all results were not new, with the method being different from the other means by which such results were obtained, it would be still a worthy contribution to the field. Furthermore, for the community at large trying to understand the importance of particular positions in Spike, knowing that a particular position identified here was also identified by works X, Y, Z adds a lot of to the field. I can only think that the authors may imagine that if one of their CAPs was identified by other means previously, it takes away from the merit of their work, but it is actually the opposite. I urge the authors to not brush away this. In fact, more important than any methodological aspect of the present work, this strengthening of evidence for particular positions by several independent methods is the most important evidence that the authors can contribute to the field.

---

## [Referee Report · Reviewer #3 (Public review)]

Summary:

The manuscript uses a combination of evolutionary approaches and structural/dynamics observations to provide mechanistic insights in the adaptation of the Spike protein during the evolution of SARS-COV-2 variants. The conclusion that CAP sites should be taken in particular account when considering the impact of the emergence of new strains and mutations is warranted.

Strengths:

The results presented in this work are very well outlined with well-written text, pleasant and well-described pictures, didactical and clear description of the methods, e.g. the discussion of how the MD equilibration procedure is applied and evaluated is clear and well argument.

The citation of relevant similar results with different approaches strengthens the reasoning; in particular, comparing the calculated scores with previous experimentally obtained data is one of the strongest points of the manuscript.

Weaknesses:

There are no replicas of the molecular dynamics (MD) simulations, understandable since it's not a MD-focused paper. However, the comparison of multiple replicas could enhance the reliability of the findings.

---

## [Author Response]

The following is the authors’ response to the original reviews.

We would like to extend our gratitude to the reviewers for their meticulous analysis and constructive feedback on our manuscript. We have revised our paper based on the suggestions regarding supporting literature and the theory behind CAPs along with detailed insights regarding our methods. Their suggestions have been extremely useful in strengthening the clarity and rigor of our manuscript.

**Reviewer #1 (Recommendations For The Authors):**
(1) There are no obvious problems with this paper and it is relatively straightforward. There are some challenges that I would like to suggest. These variants have multiple mutations, so it would be interesting if you could drill down to find out which mutation is the most important for the collective changes reported here. I would like to see a sequence alignment of these variants, perhaps in the supplemental material, just to get some indication of the extent of mutations involved.

Finding the most important mutation within a set is a tricky question, as each mutation changes the way future mutations will affect function due to epistasis. Indeed, this is what we aim to explore in this work. To illustrate this point, we included a new supplementary figure S5A. Three critical mutations that emerged quickly, and were frequently observed in other dominant variants, were S477N, T478K, and N501Y. Thus, we computed the EpiScore values of these three mutations, with several critical residues contributing to hACE2 binding. The EpiScore distribution indicates that residues 477, 478, and 501 have strong epistatic (i.e., non-additive) interactions, as indicated by EpiScore values above 2.0.

To further investigate these epistatic interactions, we first conducted MD simulations and computed the DFI profile of these three single mutants. We analyzed how different the DFI scores of the hACE2 binding interface residues of the RBD are, across three single mutants with Omicron, Delta, and Omicron XBB variants (Fig S5B). Fig S5B shows how mutations at these particular sites affect the binding interface DFI in various backgrounds, as the three mutations are also observed in the Omicron, XBB, and XBB 1.5 variants. If the difference in the DFI profile of the mutant and the given variant is close to 0, then we could safely state that this mutation affected the variant the most. However, what we observe is quite the opposite: the DFI profile of the mutation is significantly different in different variant backgrounds. While these mutations may change overall behavior, their individual contributions to overall function are more difficult to pin down because overall function is dependent on the non-additive interactions between many different residues.

**Author response image 1. sa3fig1:** (A) Three critical mutations that emerged quickly, and were frequently observed in other dominant variants, were S477N, T478K, and N501Y. EpiScores of sites 477, 478, and 501 with one another are shown with k = the binding interface of the open chain. These residues are highly epistatic, producing higher responses than expected when perturbed together. (B) The difference in the dynamic flexibility profiles between the single mutants and the most common variants for the hACE2 binding residues of the RBD. DFI profiles exhibit significant variation from zero, and also show different flexibility in each background variant, highlighting the critical non-additive interactions of the other mutation in the given background variant. Thus, these three critical mutations, impacting binding affinity, do not solely contribute to the binding. There are epistatic interactions with the other mutations in VOCs that shape the dynamics of the binding interface to modulate binding affinity with hACE2.

As we discussed above, while the epistatic interactions are crucial and the collective impact of the mutations shape the mutational landscape of the spike protein, we would like note that mutation S486P is one of the critical mutations we identify, modulating both antibody and hACE2 binding and our analysis reveals the strong non-additive interactions with the other mutational sites. This mutational site appears in both XBB1.5 and earlier Omicron strains which highlights its importance in functional evolution of the spike protein. CAPs 346R, 486F, and 498Q also may be important, as they have a high EpiScore, indicating critical epistatic interaction with many mutation sites.

Regarding to the suggestion about presenting the alignment of the different variants, we have attached a mutation table, highlighting the mutated residues for each strain compared to the reference sequence as supplemental Figure S1 along with the full alignment file.

(2) Also, I am wondering if it would be possible to insert some of these flexibilities and their correlations directly into the elastic network models to enable a simpler interpretation of these results. I realize this is beyond the scope of the present work, but such an effort might help in understanding these relatively complex effects.

This is great suggestion. A similar analysis has been performed for different proteins by Mcleash (See doi: 10.1016/j.bpj.2015.08.009) by modulating the spring constants of specific position to alter specific flexibility and evaluate change in elastic free energy to identify critical mutation (in particular, allosteric mutation) sites. We will be happy to pursue this as future work.

Minor(3) 1 typo on line 443 - should be binding instead of biding.

Fixed, thanks for spotting that.

(4) The two shades of blue in Fig. 4B were not distinguishable in my version.

To fix this, we have changed the overlapping residues between Delta and Omicron to a higher contrast shade of blue.

(5) Compensatory is often used in an entirely different way - additional mutations that help to recover native function in the presence of a deleterious mutation.

Although our previous study (Ose et al. 2022, Biophysical Journal) shows that compensatory mutations were generally additive, the two ideas are not one and the same. We thank the reviewer for pointing this out. Therefore, to clarify, we have now described our results in terms of dynamic additivity, rather than compensation.

**Reviewer #2 (Recommendations For The Authors):**
(1) The authors note that the identified CAPs overlap with those of others (Cagliani et al. 2020; Singh and Yi 2021; Starr, Zepeda, et al. 2022). In itself, this merits a deeper discussion and explicit indication of which positions are not identified. However, there is one point that I believe may represent a fundamental flaw in this study in that the calculation of EP from the alignment of S proteins ignores entirely the differences in the interacting interface with which S for different coronaviruses in the alignment interact in the different receptors in each host species. This may be the reason why so many "CAPs" are in the RBD. The authors should at the very least make a convincing case of why they are not simply detecting constraints imposed by the different interacting partners, at least in the case of positions within the RBD interface with ACE2. Another point that the authors should discuss is that ACE2 is not the only receptor that facilitates infection, TMPRSS2 and possibly others have been identified as well. The results should be discussed in light of this.

To begin with, we have now explicitly noted (on line 135) that “sites 478, 486, 498, and 681 have already been implicated in SARS-CoV-2 evolution, leaving the remaining 11 CAPs as undiscovered candidate sites for adaptation.” Evolutionary analyses are done using orthologous protein sequences, so there is no way to integrate information on different receptors in each host species in the calculation of EPs. However, we appreciate that the preponderance of CAPs in the RBD is likely due to different binding environments. We have added the following text (on line 83) to clarify our point: “Adaptation in this case means a virus which can successfully infect human hosts. As CAPs are unexpected polymorphisms under neutral theory, their existence implies a non-neutral effect. This can come in the form of functional changes (Liu et al. 2016) or compensation for functional changes (Ose et al. 2022). Therefore, we suspect that these CAPs, being unexpected changes from coronaviruses across other host species with different binding substrates, may be partially responsible for the functional change of allowing human infection.” This hypothesis is supported by the overlap of CAPs we identified with the positions identified in other studies (e.g., 478, 486, 498, and 681). Binding to TMPRSS2 and other substrates are also covered by this analysis as it is a measure of overall evolutionary fitness, rather than binding to any specific substrate. Our paper does focus on discussing hACE2 binding and mentions furin cleavage, but indeed lacks discussion on the role of TMPRSS2. We have added the following text to line 157: “Another host cell protease, TMPRSS2, facilitates viral attachment to the surface of target cells upon binding either to sites Arg815/Ser816, or Arg685/Ser686 which overlaps with the furin cleavage site 676-689, further emphasizing the importance of this area (Hoffmann et al. 2020b; Fraser et al. 2022).”

(2) Turning now to the computational methods utilized to study dynamics, I have serious reservations about the novelty of the results as well as the validity of the methodology. First of all, the authors mention the work of Teruel et al. (PLOS Comp Bio 2021) in an extremely superficial fashion and do not mention at all a second manuscript by Teruel et al. (Biorxiv 2021.12.14.472622 (2021)). However, the work by Teruel et al. identifies positions and specific mutations that affect the dynamics of S and the evolution of the SARS-CoV-2 virus in light of immune escape, ACE2 binding, and open and closed state dynamics. The specific differences in approach should be noted but the results specifically should be compared. This omission is evident throughout the manuscript. Several other groups have also published on the use of nomal-mode analysis methods to understand the Spike protein, among them Verkhivker et al., Zhou et al., Majumder et al., etc.

Thank you for your suggestions. Upon further examination of the listed papers, we have added citations to other groups employing similar methods. However, it's worth noting that the results of Teruel et al.'s studies are generally not directly comparable to our own. Particularly, they examine specific individual mutations and overall dynamical signatures associated with them, whereas our results are always considered in the context of epistasis and joint effects with CAPs, and all mutations belong to the common variants. Although important mutations may be highlighted in both cases, it is for very different reasons. Nevertheless, we provide a more detailed mention of the results of both studies. See lines 178, 255, and 393.

(3) The last concern that I have is with respect to the methodology. The dynamic couplings and the derived index (DCI) are entirely based on the use of the elastic network model presented which is strictly sequence-agnostic. Only C-alpha positions are taken into consideration and no information about the side-chain is considered in any manner. Of course, the specific sequence of a protein will affect the unique placement of C-alpha atoms (i.e., mutations affect structure), therefore even ANM or ENM can to some extent predict the effect of mutations in as much as these have an effect on the structure, either experimentally determined or correctly and even incorrectly modelled. However, such an approach needs to be discussed in far deeper detail when it comes to positions on the surface of a protein such that the reader can gauge if the observed effects are the result of modelling errors.

We would like to clarify that most of our results do not involve simulations of different variants, but rather how characteristic mutation sites for those variants contribute to overall dynamics. For the full spike, we operate on only two simulations: open and closed. When we do analyze different variants, starting on line 438, the observed difference does not come from the structure, but from the covariance matrix obtained from molecular dynamics (MD) simulations, which are sensitive to single amino acid changes.

**Reviewer #3 (Recommendations For The Authors):**
(1) On line 99 there is a misspelling, 'withing'.

It has been fixed. Thanks for spotting that.

(2) Some graphical suggestions to make the figures easier to read:In Figure 1C, a labeled circle around the important sites, the receptor binding domain, and the Furin cleavage site, would help the reader orient themselves. Moreover, it would make clear which CAPs are NOT in the noteworthy sites described in the text.

Good idea. We have added transparent spheres and labels to show hACE2 binding sites and Furin cleavage sites.

In Figure 2C the colors are a bit low contrast; moreover, there are multiple text sizes on the same figure which should perhaps be avoided to ensure legibility.

We have made yellow brighter and standardized font sizes.

Figure 3 is a bit dry, perhaps indicating in which bins the 'interesting' sites could be informative.

Thank you for the suggestion, but the overall goal of Figure 3 is to illustrate that the mutational landscape is governed by the equilibrium dynamics in which flexible sites undergo more mutations during the evolution of the CoV2 spike protein. Therefore, adding additional positional information may complicate our message.

Figure 4, the previous suggestions about readability apply.

We ensured same sized text and higher contrast colors.

Figure 5B, the residue labels are too small.

We increased the font size of the residue labels.

In Figure 8 maybe adding Delta to let the reader orient themselves would be helpful to the discussion.

Unfortunately, there is no single work that has experimentally quantified binding affinities towards hACE2 for all the variants. When we conducted the same analysis for the Delta variant in Figure 8, the experimental values were obtained from a different source (doi: 10.1016/j.cell.2022.01.001) and the values were significantly different from the experimental work we used for Omicron (Yue et al. 2023). When we could adjust based on the difference in experimentally measured binding affinity values of the original Wuhan strain in these two separate studies, we observed a similar correlation, as seen below. However, we think this might not be a proper representation. Therefore, we chose to keep the original figure.

**Author response image 2. sa3fig2:** The %DFI calculations for variants Delta, Omicron, XBB, and XBB 1. 5. (A) %DFI profile of the variants are plotted in the same panel. The grey shaded areas and dashed lines indicate the ACE2 binding regions, whereas the red dashed lines show the antibody binding residues. (B) The sum of %DFI values of RBD-hACE2 interface residues. The trend of total %DFI with the log of Kd values overlaps with the one seen with the experiments. (C) The RBD antibody binding residues are used to calculate the sum of %DFI. The ranking captured with the total %DFI agrees with the susceptibility fold reduction values from the experiments.

(3) Replicas of the MD simulations would make the conclusions stronger in my opinion.

We ran a 1µs long simulation and performed convergence analysis for the MD simulations using the prior work (Sawle L, Ghosh K. 2016.) More importantly, we also evaluated the statistical significance of computed DFI values as explained in detail below (Please see the answer to question 3 of Reviewer #3 (Public Review):)

**Reviewer #3 (Public Review):**
(1) A longer discussion of how the 19 orthologous coronavirus sequences were chosen would be helpful, as the rest of the paper hinges on this initial choice.

The following explanation has been added on line 114: EP scores of the amino acid variants of the S protein were obtained using a Maximum Likelihood phylogeny (Kumar et al. 2018) built from 19 orthologous coronavirus sequences. Sequences were selected by examining available non-human sequences with a sequence identity of 70% or above to the human SARS CoV-2’s S protein sequence. This cutoff allows for divergence over evolutionary history such that each amino acid position had ample time to experience purifying selection, whilst limiting ourselves to closely related coronaviruses. (Figure 1A).

(2) The 'reasonable similarity' with previously published data is not well defined, nor there was any comment about some of the residues analyzed (namely 417-484). We have revised this part of the manuscript and add to the revised version.

We removed the line about reasonable similarity as it was vague, added a line about residues 417-484, and revised the text accordingly, starting on line 354.

(3) There seem to be no replicas of the MD simulations, nor a discussion of the convergence of these simulations. A more detailed description of the equilibration and production schemes used in MD would be helpful. Moreover, there is no discussion of how the equilibration procedure is evaluated, in particular for non-experts this would be helpful in judging the reliability of the procedure.

We opted for a single, extended equilibrium simulation to comprehensively explore the longterm behavior of the system. Given the specific nature of our investigation and resource constraints, a well-converged, prolonged simulation was deemed a practical and scientifically valid approach, providing a thorough understanding of the system's dynamics. (doi: 10.33011/livecoms.1.1.5957, https://doi.org/10.1146/annurev-biophys-042910-155255 )

We updated our methods section starting on line 605 with extended information about the MD simulations and the converge criteria for the equilibrium simulations. We also added a section that explains our analysis to check statistical significance of obtained DFI values.